# Unveil the Secret of the Bacteria and Phage Arms Race

**DOI:** 10.3390/ijms24054363

**Published:** 2023-02-22

**Authors:** Yuer Wang, Huahao Fan, Yigang Tong

**Affiliations:** 1College of Life Science and Technology, Beijing University of Chemical Technology, Beijing 100029, China; 2Beijing Advanced Innovation Center for Soft Matter Science and Engineering (BAIC-SM), Beijing University of Chemical Technology, Beijing 100029, China

**Keywords:** bacteria, phages, restricting modification systems, CRISPR-Cas systems, aborting infection, quorum sensing

## Abstract

Bacteria have developed different mechanisms to defend against phages, such as preventing phages from being adsorbed on the surface of host bacteria; through the superinfection exclusion (Sie) block of phage’s nucleic acid injection; by restricting modification (R-M) systems, CRISPR-Cas, aborting infection (Abi) and other defense systems to interfere with the replication of phage genes in the host; through the quorum sensing (QS) enhancement of phage’s resistant effect. At the same time, phages have also evolved a variety of counter-defense strategies, such as degrading extracellular polymeric substances (EPS) that mask receptors or recognize new receptors, thereby regaining the ability to adsorb host cells; modifying its own genes to prevent the R-M systems from recognizing phage genes or evolving proteins that can inhibit the R-M complex; through the gene mutation itself, building nucleus-like compartments or evolving anti-CRISPR (Acr) proteins to resist CRISPR-Cas systems; and by producing antirepressors or blocking the combination of autoinducers (AIs) and its receptors to suppress the QS. The arms race between bacteria and phages is conducive to the coevolution between bacteria and phages. This review details bacterial anti-phage strategies and anti-defense strategies of phages and will provide basic theoretical support for phage therapy while deeply understanding the interaction mechanism between bacteria and phages.

## 1. Introduction

Bacteria and bacteriophages (phages for short) have been engaged in a constant and repeated arms race and have coexisted steadily for billions of years. Phages outnumber bacteria by ten to one and are widely recognized as the most diverse of microbes. Phages can be divided into temperate phages and lytic phages according to their intracellular proliferation modes [1]. After infecting the host, temperate phages do not lysis the host cell but remain in the lysogenic state, integrating the phage genome into the host genome, becoming prophage and replicating with the host replication. Lytic phages can cause lysis and death of the host [2]. The process of lytic phages infecting the host includes five parts: adsorption, injection, biosynthesis, assembly and release. Under certain conditions, temperate phages can enter the lysis cycle from the lysogenic state, causing the host cell to the lysis and die [3].

Bacteria have developed various strategies to resist phage infection, such as preventing phages from adsorbed on the surface of bacteria [4]; through the superinfection exclusion (Sie) block of phage’s nucleic acid injection [5]; by restricting modification (R-M) systems [6], CRISPR-Cas [7], aborting infection (Abi) [8] and other defense systems to interfere with the replication of phage genes in the host; and through quorum sensing (QS) enhancement of phage’s resistant effect [9]. At the same time, phages have also evolved a variety of counter-defense strategies, such as degrading extracellular polymeric substances (EPS) that mask receptors [10] or recognizing new receptors [11], thereby regaining the ability to adsorb host cells; modifying its genes to prevent the R-M systems from recognizing phage genes [12] or evolving proteins that can inhibit the R-M complex [13]; through mutate the target sequence [14], build nucleus-like compartments [15] or evolved anti-CRISPR (Acr) proteins [16] to resist CRISPR-Cas systems; and by producing antirepressors [17] or blocking the combination of autoinducers (AIs) and its receptors [18] to suppress the QS (Table 1).

The competition between bacteria and phages has never stopped, and the rapid co-evolution between them has a positive impact on improving the evolution rate of phages and bacteria [19]. Understanding these adversarial strategies is of great significance to the research fields of practical applications.

## 2. Bacterial Anti-Phage Strategies

### 2.1. Blocking Phage Adsorption

Preventing phages from attaching to host cells is the first step in the bacterial defense battle. Strategies to avoid phage adsorption can be divided into two categories: (i) loss or structural change of receptors and (ii) construction of physical barriers to phage infection. The absence of receptors is a key part of bacterial resistance to phage infection [20] (Figure 1a).

Through gene mutation or deletion, the receptor cannot be expressed or the structure of the receptor can be changed to protect bacteria from phage infection. For example, phage OWB uses tail tubular proteins A and B to recognize the transmembrane protein encoded by *Vibrio parahaemolyticus vp0980*. However, due to the lack of phage-recognized receptors, the mutant hindered the adsorption of phage OWB [4]. *Listeria monocytogenes* serovar 4b is mutated from serotype 4b to the more virulent serotype 4d by mutating the gene encoding the glycosylation of teichoic acid. Serotype 4d is resistant to infection by phages with wall-teichoic acids as recognition receptors due to the absence of galactose from wall-teichoic acids and lipoteichoic acid molecules [21].

The cell wall is usually thought to protect bacteria from environmental threats [22]. However, because the cell wall contains receptors that the phages can recognize, bacteria with cell wall defects can resist phage adsorption in some cases. Cell wall deficiency is one of the mechanisms of bacteria hindering phage adsorption [23]. By losing or modifying the receptors of jumbo phages, such as lipopolysaccharide and type IV pili, the adsorption of phages to *Pseudomonas aeruginosa* (*P. aeruginosa*) PA5oc can be inhibited [24]. At the same time, the virulence and pathogenicity of the bacteria are reduced due to the reduction in the virulence factors such as lipopolysaccharide and type IV pili. In addition, studies have shown that some phage-resistant mutant strains can also affect the growth of bacterial biofilm. The PA1S_08510 gene of *P. aeruginosa* PA1 strains encodes the O-antigen polymerase Wzy. The phage-resistant mutant strain PA1RG hinders the infection of phage PaP1 which uses O-antigen as a receptor due to the lack of O-antigen on its surface [25]. Meanwhile, the reduction in biofilm in the mutant strain PA1RG can lead to the re-sensitivity of drug-resistant bacteria to some antibiotics.

The capsule (K antigen), which acts as a virulence factor, can also act as a receptor for some phages. Phages ΦFG02 and ΦCO01 are able to infect *Acinetobacter baumannii* (*A. baumannii*) strains AB900 and A9844, respectively, using the capsule as receptors. The phage-resistant mutants ΦFG02-RAB900 and ΦCO01-RA9844 affected the genes responsible for capsule biosynthesis, *gtr29* and *gpi*, respectively, through a single nucleotide deletion at the K locus. The deletion of the capsule as phage receptors, ΦFG02-RAB900 and ΦCO01-RA9844, results in the interruption of phage adsorption [26]. Phage Phab24 also uses the capsule of *A. baumannii* as receptors and the outer membrane as secondary receptors [27]. Besides, both studies have found that phage-resistant mutants lacking bacterial capsules can be re-sensitized to antibiotics, which is conducive to the study of phage therapy and phage therapy in combination with antibiotics for the treatment of drug-resistant bacteria.

Masking phage receptors by physical barriers such as outer membrane vesicles (OMVs) or EPS can also prevent phages from being adsorbed to bacterial surfaces. Composed of polysaccharides, proteins and nucleic acids, EPS not only enables bacteria to survive in harsh environments but is also significant in fighting against attacks on bacteria by phages and antibiotics [28,29]. The outer membrane protein OmpA of *Escherichia coli* (*E. coli*) has been proven to be the receptor of some T-even-like phages. The outer membrane lipoprotein TraT interacts with OmpA to inhibit OmpA-specific phages binding to the receptors and thus inhibit phage adsorption [30]. The cell-binding protein A of *Staphylococcus aureus* (*S. aureus*) can also mask teichoic acid which acts as the phage receptor and achieve inhibition of adsorption [31]. There are various types of OMVs secreted by bacteria, which have functions such as transporting virulence factors and bacterial communication [32]. In addition, OMVs can also act as a protective umbrella for bacteria, hindering phage’s adsorption to bacterial surface receptors. The phage infection can be effectively avoided by allowing phages to attach to OMVs [33].

### 2.2. Blocking Phage’s Injection

Blocking the injection of phage nucleic acids through Sie is the second line of defense established by bacteria. Sie exists widely in animal [34] and plant viruses [35]. Sie prevents the host cell infected by the temperate phages from being infected again by identical or highly similar phages. Various proteins encoded by phages establish the Sie mechanism by (i) inhibiting phages from binding to receptors, (ii) blocking phage’s DNA injection, and (iii) inhibiting phage tail tube penetration of the plasma membrane [36].

The protein gp05 encoded by temperate phage D3112 is a twitching inhibitory protein that affects the assembly of type IV pilus tail fiber proteins (TFPs) by interacting with the bacterial type IV pilus assembly protein PilB. Thus, preventing phage MP22, which also uses *P. aeruginosa* TFPs as receptors, from re-infecting the host cell [37] (Figure 2). T-even phages can establish the Sie mechanism after infecting bacteria. For example, when other T-even phages re-infect the host, the T4 phage encodes two proteins, Imm and Spackle, that enable about 50% of the DNA to remain in the head of the phage and the rest to be degraded by endonuclease I in the peripheral space of the host cell, thus preventing the injection of the phage’s DNA into the bacteria [5]. Among them, Imm blocks DNA transfer into the cytoplasm by binding to the receptor on the inner membrane. Spackle blocks the degradation of peptidoglycan. The early phage gene product Sp inhibits the activity of gp5 by forming the Sp-gp5 complex with the phage tail spike protein gp5, hindering the local degradation of host cell peptidoglycan by the lysozyme, and preventing the phage’s tail from entering the cytoplasm. In addition, studies have shown that Sp does not interact with T4 endolysin. On the one hand, the structure of endolysin is different from that of gp5; on the other hand, as a late gene product, endolysin seems more difficult to interact with Sp [38]. Since lytic activity in T4 is not significantly inhibited, host cells containing prophages can still be lysed and release progeny phage particles. Sie has been extensively studied in double-stranded DNA (dsDNA) phages, while single-stranded DNA (ssDNA) phages belonging to the family *Microviridae* have also been proven to block a phage’s DNA injection through the highly variable region of the DNA pilot protein, thus preventing the repeated infection of other microviruses [39].

The antisense RNA of some phages can regulate the expression of Sie-related genes, thus affecting the mechanism of Sie action. In the lysogenic state, the gene *sieB* of temperate phage P22 encodes two peptides, namely, the exclusion protein SieB and Esc. The exclusion protein SieB mediated the Sie mechanism, and Esc inhibited SieB. However, since the antisense RNA synthesized by phage P22 has an inhibitory effect on SieB synthesis, and the regulatory gene *sieB* selectively expresses Esc, P22 still has a chance to re-infect the host cell, thus escaping the Sie mechanism [40].

### 2.3. Interfering with Phage Replication

Blocking the replication of phage nucleic acid in host cells is the third line of defense that bacteria have established. Currently, well-studied defense systems include (i) R-M systems, (ii) CRISPR-Cas adaptive immune systems, and (iii) Abi. In addition, some emerging defense systems are being discovered. Here, several anti-phage defense systems widely distributed in bacteria are introduced in detail. At the same time, the newly discovered defense systems in recent years are briefly listed to enrich the understanding of the mechanism of bacterial anti-phage action.

#### 2.3.1. Restriction-Modification Systems

R-M systems are the classical defense system that interferes with the replication of phages in the host, which generally acts in the early stage of phage infection. R-M systems can be classified into four classes (I–IV) [6], among which, type II R-M systems have been the most widely studied. The type I–III R-M system consists of genes encoding restriction endonuclease (REase) and methyltransferase (MTase), while the type IV R-M system contains only REase-related genes. MTase methylates its own DNA recognition site to distinguish unmodified foreign DNA, and REase cleaves the phosphodiester bond of unmethylated foreign phage DNA (Figure 3a). In addition, R-M systems rely on mobile genetic elements to promote bacterial genome evolution through horizontal gene transfer [41,42]. The REase and MTase of the type I R-M system are composed of three subunits, HsdR (DNA-restricted translocation subunit), HsdM (DNA-modified subunit), and HsdS (DNA-specific subunit) encoded by the host-specific determining factor (*hsd*) gene [43], while the REase of type II R-M system is composed of a single subunit.

However, sometimes the R-M systems can lead to the cleaving of its DNA due to false recognition, thus causing autoimmunity [44]. In addition, when the expression of REase and MTase is unbalanced, it can be fatal. When the REase/MTase ratio is too high, the host’s DNA may be cleaved by REase before it is methylated. In contrast, when the REase/MTase ratio is too low, phage DNA is modified by MTase and cannot be cleaved by REase, resulting in the host being infected. After the transformation of *E. coli* cells with plasmids carrying the Esp1396I type II R-M system, MTase is first synthesized to rapidly modify bacteria genome in order to avoid cleavage of the host DNA by the synthesized REase [45]. On the one hand, the promoter of MTase gene transcription is stronger than that of REase; on the other hand, because type II REase needs to be active in the form of homologous dimer or homologous tetramer, low concentration of gene expression products will limit the formation of polymerization.

At present, one of the regulatory mechanisms of type II R-M system expression is dependent on the C protein. Transcription factor C protein in C-dependent R-M systems can regulate the expression levels of REase and MTase and may play different regulatory roles in different R-M systems [46]. The binding site of the C protein, C-box, consists of a pair of reverse repeats that form a negative feedback loop with the C protein homologous dimer. When the C protein concentration is low, the expression of downstream REase is activated. On the contrary, REase expression was inhibited when C protein concentration was high. Although a low concentration of C protein can activate downstream REase expression, the tandem REase promoter still plays a dominant role in the expression of REase gene [47]. C protein is conducive to the expression of MTase before REase, and the absence of the C gene will lead to the premature expression of REase, resulting in REase cleaving of its DNA [48].

#### 2.3.2. CRISPR-Cas Adaptive Immune Systems

The CRISPR-Cas systems defend against phages by recognizing and cleaving phage genes [49]. Moreover, some CRISPR-Cas systems can still exert immunity against methylated phage DNA [50]. CRISPR-Cas are widespread in bacteria, such as the type I-E CRISPR-Cas system found in *E. coli* K12 [7], the type II CRISPR-Cas system found in *Streptococcus agalactiae* (*S. agalactiae*) strain GD201008-001 [51], the type I-C CRISPR-Cas system found in *Actinomycetes Eggerthella lenta* [52]. CRISPR-Cas systems are of great significance for the evolution of bacteria and for enhancing the adaptability of bacteria to the environment [53].

The process of CRISPR-Cas systems to play an immune role can be divided into three steps: adaptation (acquisition of foreign genes), expression (transcription of CRISPR array, maturation of transcripts and formation of effector complexes), and interference (targeting and cleavage of foreign genes) (Figure 4a). Spacers of the CRISPR-Cas systems are generally obtained at the early stage of phage genes injection into host cells [54]. The CRISPR-Cas systems can insert phage genes (protospacer) into CRISPR sites on the host genome. Different phages protospacer inserts into the bacterial genome constitute rich and diverse spacers in the host genome. The more types of phages that infect the host and the more diverse the spacers, the more beneficial it is for bacteria to defend against infection by different phages [55]. In addition to Cas proteins, the trans-activating CRISPR RNA (tracrRNA), which is partially complementary to crRNA, is also essential in the processing of CRISPR-derived RNA (crRNA) from precursor crRNA (pre-crRNA) [56]. The crRNA and tracrRNA are called guide RNA (gRNA). The gRNA and Cas proteins form the effector complex to target and cleave phage nucleic acids. TracrRNA has only been found in type II and V-B systems, and effector complexes of other CRISPR-Cas systems consist only of crRNA with Cas proteins.

The protospacer adjacent motif (PAM) of the CRISPR-Cas systems is located near the protospacer. On the one hand, PAM can assist the Cas protein to recognize foreign genes more accurately and avoid causing host autoimmunity. On the other hand, PAM also provides the possibility for phage point mutations to escape the CRISPR-Cas systems. In addition, crRNA terminal sequences can distinguish host genes by recognizing repeats (about 8 bp) in CRISPR sequences, thereby protecting host genes from cleavage [57]. Nevertheless, the phenomenon of self-targeting spacers still occurs sometimes, and the CRISPR-Cas system mistakenly targets and cuts its own genome, leading to the occurrence of autoimmunity [58].

According to the composition of Cas proteins in effector complexes, CRISPR-Cas systems can be classified into class 1 systems (consistsing of multiple Cas proteins) and class 2 systems (consisting of a single Cas protein) [59]. Each type is further divided into multiple subtypes. Different Cas proteins play different roles, such as the integration of foreign DNA, the maturation of crRNA and the cleavage of foreign genes [60,61]. Cas12a protein in type V CRISPR-Cas acts as the RNase to process pre-crRNA into crRNA, and Cas13 in type VI CRISPR-Cas can play the function of targeting and cleaving single-stranded RNA (ssRNA) [62]. Some Cas proteins play a single role, while some Cas proteins can have multiple functions at the same time. For example, Cas9 plays an indispensable role in all stages of the type II CRISPR-Cas system defense against phages [63].

Obviously, the more phage species, the more abundant and diverse spacers in the host genome, which is more conducive to the evolution of bacterial CRISPR-Cas systems. Additionally, the higher the content of CRISPR spacers, the more sensitive the bacteria to phages [64]. The increase in phage abundance is conducive to the increase in CRISPR-Cas systems abundance. Additionally, when the abundance of phages is constant, the phage species is inversely correlated with the abundance of CRISPR-Cas systems [65]. In addition to phage diversity, bacterial species richness can also have an impact on the evolution of CRISPR-Cas systems. When there are infectious phages in the environment, the more intense the competition between bacterial species is and the more beneficial the anti-phage evolution of bacterial CRISPR-Cas is [66]. Moreover, when there are abundant bacterial species and strong interspecific competition in the environment, bacteria will preferentially adopt CRISPR-Cas rather than surface receptor mutation to defend against phage infection [66]. The acquisition rate of spacers has also been shown to be one of the decisive factors affecting the abundance of spacers. The faster CRISPR-Cas can obtain spacers, the more conducive it is to increaseing the diversity of spacers [67]. In addition, the speed of phage development is also one of the important factors affecting the evolution of bacterial CRISPR-Cas systems. When phages develop too fast in the host cell, it is not conducive for the host to acquire spacers [68]. When the host growth is stagnant, the growth rate of phages in the host is delayed, but the acquisition process of spacers is not affected at this time. Therefore, when the host cell growth is inhibited by the external environment and thus inhibits the speed of phage growth, the evolution of the CRISPR-Cas systems may be beneficial.

#### 2.3.3. Abortive Infection

The Abi immune systems are activated during the middle and late stages of phage maturation. By inhibiting their own metabolism, bacteria lead to their own growth arrest and eventually lead to the death of bacteria, thus avoiding the maturation and release of phages. Although Abi can prevent other bacteria from being infected by mature progeny phages, it is at the cost of host cell suicide, so Abi can also be considered as the last line of defense of the bacterial anti-phage mechanism. Abi can impede phage replication based on (i) CRISPR-Cas systems (Figure 5a), (ii) toxin-antitoxin (TA) systems (Figure 5b), and (iii) cyclic oligonucleotide-based anti-phage signaling system (CBASS) [8] (Figure 5c). This part will introduce the Abi immune systems from three aspects: CRISPR-Cas, TA and CBASS.

When phages infect host cells and CRISPR-Cas fails to provide good protection to bacteria, they may choose to mediate Abi to protect other uninfected cells by sacrificing themselves [69]. For example, type I-F CRISPR-Cas of *Pectobacterium atrosepticum* inhibits the maturation of the lytic phages ΦTE and ΦM1 by mediating Abi [70]. This may be caused by the indiscriminate cleavage of CRISPR-Cas’s own genes, but the specific mechanism of Abi caused by type I-F CRISPR-Cas is still unclear. For example, both Cas14 and Cas12 with a single RuvC domain can nonspecifically target ssDNA [71], and Cas13 can nonspecifically target ssRNA. In fact, the Cas protein with endonuclease activity, which nonspecifically targets phages and bacterial genes, is thought to be one of the causes of CRISPR-Cas mediated Abi.

The TA systems can also mediate the Abi immune systems and block phage infection. TA systems also have a variety of functions such as controlling bacterial growth, biofilm formation, maintaining genome stability, and dormancy [72]. This part mainly introduces its function of resisting phage infection. TA systems are widespread in bacteria and consist of toxins and less stable antitoxins. Among them, toxins usually play the role of inhibiting bacterial growth in the form of protein, and antitoxins play the role of inhibiting toxins in the form of protein or RNA [73]. TA systems are divided into six types (Type I–Type VI. The phage activation mechanism of TA systems is not fully clear. At present, a relatively clear molecular mechanism is considered [74]: under normal circumstances, antitoxins can play a role in inhibiting toxins. At this point, the toxin is neutralized, and the TA systems are not activated. However, when phages infect bacteria, the TA transcription function of bacteria is hindered, and unstable antitoxin is degraded before toxin, resulting in toxin accumulation, which leads to bacterial growth arrest or death.

Type I TA system toxins (<60 aa) can hinder the synthesis of ATP or act as a nonspecific endonuclease to cleave methylated and unmethylated DNA without distinction, thereby retarding bacterial growth or causing bacterial death [75]. For example, the toxin RalR in the type I TA system of *E. coli* mediates bacterial death by indiscriminate DNA cleavage, while the antitoxin RalA blocks the translation of toxin proteins by complementary pairing with the mRNA guiding RalR synthesis [76]. The type I TA system Hok/Sok can inhibit phage T4 infection. Phage T4 can block the host transcription process and the antitoxin Sok is not stable. As a result, Sok is degraded first and the toxin Hok is activated, thus inhibiting the growth of host cells [77].

Antitoxins forming complexes with toxins is one of the common ways in the type II TA system that toxins mediate Abi systems by cleaving mRNA as RNase to inhibit bacterial protein synthesis [78,79]. The toxin RnlA in the type II TA system has RNase activity, and the toxin RnlA exists as a homodimer with two conformations [80]. The antitoxin RnlB inhibits RnlA by binding to the HEPN (higher eukaryotes and prokaryotes nucleotide) domain of the toxin RnlA.

The antitoxin (the form of RNA) of type III TA interacts with toxins directly to inhibit the toxin [81]. Phage T4 can inhibit the transcription of toxin ToxN and antitoxin ToxI in the type III TA system after *E. coli* infection. Because antitoxin ToxI is more unstable than toxin ToxN, a large amount of ToxN is accumulated [74]. The toxin ToxN, which acts as an RNase, inhibits phage translation and maturation mainly by recognizing the GAAAU motif and cleaving the mRNA of the phages. When no phages infect bacteria, the ToxI pseudoknot with a sequence length of 36 nt interacts with three ToxNs to form a complex, thereby inhibiting the activity of ToxN [82].

The antitoxin (the form of protein) of type IV TA systems binds to the target of action of the toxin rather than forming a complex with the toxin protein. For example, in the type IV TA system of *S. agalactiae*, the antitoxin AbiEi inhibits the binding of the toxin AbiEii, which acts as a nucleotidyltransferase (NTase) to GTP [83]. The antitoxin GhoS of the type V TA system can specifically recognize and cleave the U- and A-rich sites in the mRNA of toxin GhoT, thus hindering the dissolution of the cell membrane by GhoT [84]. Different from the TA systems found in the past, in the type VI TA system, the toxin SocB, which can bind to the sliding clamp to hinder the replication process, can be degraded by the protease ClpXP with the participation of the antitoxin SocA [85]. Moreover, the type I TA system in *Clostridium difficile* (*C. difficile*) co-localizes with CRISPR arrays [86]. This means that CRISPR-Cas and TA systems may have some potential connection in mediating Abi.

In addition, CBASS is also one of the ways to mediate bacterial Abi. The CBASS immune system is widespread in bacteria and can impede phage replication. After a phage infection, cGAS/DncV-like nucleotide transferase (CD-NTases) in bacterial CBASS can synthesize second messengers such as cyclic dinucleotides and cyclic trinucleotides [87]. The CD-NTase-related protein Cap, which acts as an effector protein, is activated after binding to specific second messengers and induces cell death by disrupting cell membranes, cleaving intracellular DNA, or other means. The crystal structure of the Cap protein can determine the type of the second messenger that binds to it [88]. After *Yersinia* is infected by phages, the receptor domain of oligomeric cyclic dinucleotide formed by the 8-stranded β-barrel scaffold specifically binds to signaling molecules (second messengers) to promote a bacterial inner membrane rupture and mediate bacterial death [89]. Cyclic AMP-AMP-AMP (cAAA) can also induce bacterial death after binding to the homotrimer DNA endonuclease NucC (nuclease, CD-NTase-associated) in CBASS. NucC can induce bacterial death by cleaving intracellular DNA, thereby hindering phage replication in the cell. In addition, the combination of cAAA with the triple symmetric allosteric pocket of NucC can promote the formation of NucC homohexamer, which also has DNA cleavage activity [90]. Cap4 in *Enterobacter cloacae* (*E. cloacae*) can form a SAVED domain by the fusion of two CARF (CRISPR-associated Rossman fold). After specifically binding 2′-5′- and 3′-5′ -linked cyclic oligonucleotide signals, the SAVED domain activates the dsDNA endonuclease activity of Cap4, and then cleaves intracellular DNA, thereby hindering the viral replication process [91].

CBASS can be divided into four types according to different cyclase genes, auxiliary genes (*cap*) and signaling molecules [92]. Type I CBASS (42%) had no *cap* gene and induced bacterial death by forming small holes in the bacterial membrane; type II CBASS (39%) contained *cap2* and *cap3* genes; type III CBASS contained three auxiliary genes: *cap6* (encoding TRIP13/Pch2 domains), *cap7* (encoding a single HORMA domain) and *cap8* (encoding two HORMA domains); type IV CBASS contains *cap9–11* gene, which is small in quantity and it is not clear whether type IV CBASS can resist phage infection. Among them, the HORMA domain in type III CBASS can form HORMA complexes that can synthesize signaling molecules after binding with CD-NTases [93]. Additionally, TRIP13 ATPase can exert a negative regulatory effect by decomposing the HORMA complex.

### 2.4. Quorum Sensing

The aforementioned mechanisms that hinder phage adsorption, injection and replication are all introduced from the level of individual bacteria. In addition, bacteria can also be regulated as a group through QS [94]. QS, which has the function of intercellular communication, is composed of quorum-sensing signal synthase, extracellular signaling molecules called AIs and receptors [95]. AIs synthesized by quorum-sensing signal synthase can play a role in regulating population density [96], regulating virulence factors [97] and resisting phage infection after binding with the corresponding receptor. The ways of QS resisting phage infection can be divided into two: (i) cooperating with CRISPR-Cas systems to stimulate the expression of *cas* genes [98] or promoting the identification and cleaving of target genes [9]; (ii) inhibiting phage adsorption by directly reducing phage receptors.

Under the condition of high population density, the QS system can enhance the recognition and cleavage of phage genes by CRISPR-Cas systems [99]. For example, under the condition of high population density, the QS of *P. aeruginosa* PA14 can promote the expression of *cas* gene-encoding nuclease [98]. The smaI/smaR-type QS system can also enhance the immunity of CRISPR-Cas systems to phages in Serratia under the condition of high population density [100]. The quorum sensing signal synthase smaI can synthesize the acyl-homoserine lactones (AHL) signaling molecule *N*-butanoyl-_L_-homoserine lactone (C4-HSL). When the host population density is high, C4-HSL increases. C4-HSL can bind to its receptor smaR, thus blocking the inhibition of *cas* gene by DNA-binding repressor smaR (Figure 6a). In addition, the smaI/smaR-type QS system can also promote the ability of type I-E and I-F CRISPR-Cas systems to capture spacers, thus enhancing the adaptability of CRISPR-Cas systems. Thus, at high host population densities, the facilitation of the CRISPR-Cas system by the QS system may be aimed at reducing the adaptive cost of bacterial resistance to phages.

QS can directly reduce phage adsorption receptors to hinder phage adsorption. For example, the binding of AHL in *E. coli* with its receptor SdiA can reduce phage λ receptor LamB and hinder phage adsorption [101] (Figure 6b). In addition, the QS system can also increase the biofilm of bacteria under the condition of low cell density [102] (Figure 6c). The increase in biofilms may be beneficial for masking phage receptors. However, the use of biofilms to mask receptors or to reduce the number of receptors does not seem to be the main way for the QS system to defend against phages. Under the condition of high population density, biofilm production may be inhibited rather than promoted by the QS system [103].

## 3. Anti-Defense Strategies of Phages

### 3.1. Regaining the Ability to Identify and Adsorption Host

Bacteria can block phage adsorption by disabling the expression of the receptor, changing the structure of the receptor or masking the receptor with a physical barrier formed by EPS. Given the mechanism of bacteria hindering phage adsorption, phages can (i) recognize new receptors or (ii) degrade EPS by depolymerase [10], and then acquire the ability to recognize and adsorb the host again (Figure 1b).

Receptor binding protein (RBP) J is the host-recognizing trimer protein. Phage λ originally recognized the LamB receptor of host *E. coli* by the J protein. However, when phage λ is co-cultured with *E. coli* lacking the LamB receptor, phage EvoC, which recognizes the new receptor OmpF, can be isolated [11]. Additionally, both OmpF and LamB are trimeric structures. The RBP mutation of phages is not only an anti-defense measure against bacteria but is also conducive to the expansion of the host selection range of phages [104]. In addition, phages can also degrade EPS through depolymerase and acquire the ability to recognize the receptor. Depolymerases are divided into lyases (water-free molecules after the substrate is cleaved) and hydrolases [105]. Depolymerase is widely found in phages. For example, *A. baumannii* phage IME200 can express the depolymerase Dpo48 [106], and phage IME205 can express the depolymerases Dpo42 and Dpo43 [107]. Phage Sb-1 can degrade the EPS of methicillin-resistant *S. aureus*. Furthermore, the lack of EPS protection can increase the sensitivity of bacteria to antibiotics [108]. Therefore, phage depolymerase can not only enhance the recognition ability of the host but also be used in combination with antibiotics as a new type of antibiotics [109].

### 3.2. Anti-Defense Strategies for R-M Systems

Aiming at the cleavage of unmethylated phage genes by bacterial R-M systems, phages (i) modify their own genes to block the cleavage of phages nucleic acids by REase (Figure 3b) or (ii) evolve the overcome classical restriction (Ocr) protein and the restriction of DNA A (ArdA) protein that can inhibit the R-M complex (Figure 3c).

Gene modification is one of the ways that phages resist the R-M systems. For example, phages T4gt, T4, Xp12, and SP8 modify pyrimidine in their DNA to 5-hydroxymethylcytosine (5hmC) and glucose-5-hydroxymethylcytosine, respectively. 5-methylpyrimidine and 5-hydroxymethyldeoxyuridine resist type II REase [12]. The iron-binding protein Mom produced by phage Mu can methyl carbamoylation its genes after binding to the cofactor acetyl CoA and Fe^2+/3+^, thus, it can resist various REases [110]. There is a 7-deazaguanine modifier gene cluster with a length of about 5940 bp in phage CAjan, including genes *folE*, *queD*, *queE*, *queC*, *yhhQ* and *dpdA* [111]. GTP can be converted to 7-cyano-7-deazaguanine (preQ_0_) under the catalysis of four enzymes (FolE, QueD, QueE and QueC) [112]. Phage CAjan inhibits the cleavage of phage DNA by restriction endonucleases with GA and GGC as recognition sites by modifying GTP to preQ_0_ in the specific sequences GA and GGC in phage DNA.

In addition, Ocr and ArdA produced by phages can also inhibit the type I R-M system. Both Ocr dimer and ArdA dimer bind to the gap between HsdRs subunits motor 1 and motor 2 in a manner that mimics DNA, inhibiting the binding of the R-M complex to phage DNA, but not its conformational transition [13]. The Ocr dimer expressed by phage T7 occupies the DNA binding site in the type I R-M system restriction enzyme EcoKI (R_2_M_2_S_1_) by simulating the negative surface charge of DNA (~24 bp) and the approximate 46° bending of the DNA helix axis [113,114]. The Ocr protein has also been proven to inhibit BREX defenses similar to R-M systems. The Ocr dimer expressed by phage T7 binds to the BREX system complex by simulating the shape and surface charge of DNA (~20 bp) and inhibits the BREX methylation of adenine fifth in the specific non-palindromic sequence of host DNA [115]. Furthermore, the Ocr dimer can also compete with sigma factors for nucleic acid binding channels of bacterial RNA polymerase, which inhibits sigma factors from recruiting RNA polymerase to bacterial DNA promoters, thus impeding the host transcription process [116].

### 3.3. Anti-Defense Strategies for CRISPR-Cas Systems

CRISPR-Cas adaptive immune systems are one of the most important strategies for bacteria to fight phage infection. In the face of precise attacks by bacteria, phages have also evolved a series of counterattack measures to counter the bacterial defense systems. The counterattack strategies of phages against bacterial CRISPR-Cas mainly include (i) constructing nucleus-like compartments to shield nuclease; (ii) mutating the target sequence to block the recognition of effector complexes; (iii) through Acr to inhibit the recognition or cleaving of foreign nucleic acids by effector complexes (Figure 4b).

Recent studies have proven that giant phages can produce a proteinaceous nucleus-like compartment, and this protein shell can act as a physical barrier to protect phage dsDNA from nuclease hydrolysis [15]. For example, the *Serratia* giant phage PCH45 without the *acr* gene and DNA-modifying enzyme gene can encode the nucleus-like shell to resist bacterial CRISPR-Cas systems through the *gp033* gene, and then successfully infect *Serratia* with type I-E and I-F CRISPR-Cas [117]. However, because the phage protein translation process is located in the bacterial cytoplasm, mRNA, which is not protected by the nucleus-like compartment, can be targeted for cleavage by type III and VI CRISPR-Cas systems. Therefore, the phage-constructed nucleus-like compartment cannot resist CRISPR-Cas systems of type III and VI (the type III system can target foreign DNA transcripts, and Cas13 in the type VI system can cleave ssRNA [62]). The giant phage ΦKZ without the *acr* gene can also construct the nucleus-like compartment to resist bacterial types I-C, I-F, II-A and V-A CRISPR-Cas, but not type III CRISPR-Cas systems [118]. Furthermore, the nucleus-like compartment not only antagonizes the CRISPR-Cas adaptive immune systems but also impedes the R-M systems by shielding restriction enzymes [118].

Phages can block the recognition of target genes by CRISPR-Cas systems through the DNA glycosylation or mutation of PAM/protospacer. For example, the glycosylation of 5hmC in the DNA of phage T4 by glycosyltransferase can inhibit the recognition of target genes by type I-E and II-A CRISPR-Cas systems [119]. The mutation of PAM and protospacer of phage M13 enabled the phages to successfully infect *E. coli* [14]. However, the CRISPR-Cas systems have a certain fault tolerance rate for the mutation of the target gene, and the CRISPR-Cas systems can re-resist the phages by obtaining the new spacer [120].

The Acr evolved by phages is a widely discovered and studied anti-CRISPR-Cas strategy. At present, Acrs have been found to hinder the recognition or cleavage of foreign nucleic acids by effector complexes in CRISPR-Cas systems such as type I-C, I-F, II-A, II-C, III-A, V-A and VI-B by acting alone or in combination [16].

The identified type I-F Acrs include AcrIF1-4, AcrIF7-9, AcrIF11, AcrIF14, etc. For example, AcrIF4 blocks the formation of the active conformation of the Csy1 complex by interacting with the I-F CRISPR-Cas surveillance complex (the Csy complex) [121]; AcrIF7 mimics the bases of the target sequence and occupies the binding site of the Csy complex to DNA [122]; AcrIF9 interacts with Csy3 and induces non-specific binding of the Csy complex to DNA, which makes the Csy complex lose its specific targeting ability [123]; AcrIF11 deprived the Csy complex of dsDNA binding activity [124]. Gene*35* of phage JBD30, gene*30* of phage D3112 and gene*35* of phage JBD5 encode AcrIF1, AcrIF2 and AcrIF3 with anti-I-F CRISPR-Cas function, respectively [125]. Among them, AcrIF1 and AcrIF2 bind to the Csy3 and Csy1-Csy2 heterodimers of the Csy complex, respectively. AcrIF3 affects Cas3 recognition and recruitment by directly binding to Cas3 nuclease rather than binding to the Csy complex, thereby preventing Cas3 from cleaving phage DNA.

Currently, the reported type II Acrs include AcrIIA1-2, AcrIIA4, AcrIIA14, AcrIIA22-23, AcrIIC1-4, etc. The mechanism of action of traditional AcrII can be divided into three categories: (i) interacting with the HNH domain of Cas9 (responsible for cleaving complementary chains) or the RuvC domain (responsible for cleaving non-complementary chains), (ii) competing with PAM recognition sites, and (iii) blocking sgRNA recruitment. The C-terminal domain of AcrIIA1 can interact with the HNH domain of Listeria Cas9 to induce the inactivation and degradation of Cas9 through a multi-step mechanism [126]. Both AcrIIA2 and AcrIIA4 can hinder the recognition of target genes by occupying the PAM recognition site of Cas9 [127,128]. In addition, AcrIIA4 can also interact with the RuvC domain of Cas9 to block Cas9 from cleaving target genes [129]. The C-terminal domain of AcrIIA14 binds to the HNH domain of Cas9, which can inhibit the cleavage activity of Cas9 [130]. Similarly, AcrIIC1 can also directly bind to Cas9 and inhibit the cleavage activity of Cas9 [131]. The highly negatively charged AcrIIC2 can bind to the positively charged bridge helix of Cas9. AcrIIC2 prevents gRNA from forming complexes with Cas9 by occupying the gRNA binding site on Cas9 [132]. The two AcrIIC3 links the HNH domain of Cas9 and the other REC2 domain of Cas9, causing Cas9 dimerization and hindering the formation of the active conformation of the HNH domain, thus inhibiting the binding and cleavage of Cas9 to the target gene [133,134].

In addition, AcrIII, AcrV and AcrVI have been gradually discovered. AcrIII-1 with ring nuclease activity degrades cyclic tetra-adenylate (cA_4_) into A_2_>P and A_2_-P in the form of a dimer, thereby inhibiting the activation of RNase by cA_4_ [135]. Negatively charged AcrVA1 can occupy the PAM binding site of Cas12. Moreover, after AcrVA1 binds to Cas12, crRNA can be cleaved into two parts by Cas12. Additionally, AcrVA4 dimer can block the formation of the active conformation of Cas12a [136]. Similarly, after binding to Cas13a, AcrVIA1 inhibits the formation of active conformations of Cas13a that can bind to target RNA [137]. AcrVIA2 or AcrVIA3 can bind to the Cas13a-crRNA complex instead of Cas13a, hindering the cleavage of the target RNA [138]. In addition, Chevallereau A, et al. demonstrated that when Acr-positive and Acr-negative phages co-infected bacteria, the presence of Acr-positive phages facilitated the maturation of Acr-negative phage replication in the host cell [139]. With the deepening of the research on phages Acr, more and more new Acr are being discovered. This helps people to enrich their understanding of phage’s anti-bacterial mechanisms.

Besides, the anti-CRISPR-associated (*aca*) gene, which is found near the *acr* gene, encodes the protein Aca, which contains a helix-turn helix domain. The Aca2 homodimer of *Pectobacterium carotovorum* phage ZF40 binds to the *acr* promoter and can inhibit the transcription of *acr* [140]. The *aca1* gene located downstream of the *acrIF1* gene in *P. aeruginosa* phage JBD30 can express the Aca1 protein. Aca1 binds to the *acrIF1* promoter in the form of the homologous dimer to inhibit the transcription of the *acrIF1* gene [141].

### 3.4. Anti-Defense Strategies for QS System

For the QS system of bacteria, phages’ counterattack strategies include: (i) expressing anti-repressor and binding cI repressor to make phages enter the lysis cycle (Figure 7a); (ii) synthesizing receptors that can bind to AIs or synthesizing AIs-like proteins to prevent AIs from binding to their corresponding QS receptor.

Temperate phages are essential in the evolution and diversity of microbial populations [142]. Temperate phages do not lysis host cells and produce progeny phages during the lysogenic cycle, but make their genes integrate with host bacteria chromosomes and pass along with the division of bacteria. When the phages enter the lysis cycle from the lysogenic state, they can produce mature progeny phages, which mediate the lysis death of bacteria. The switch between the phage’s lysogenic cycle and lysis cycle is the cI repressor in host cells. By binding with the Q promoter, the cI inhibitor inhibits the expression of the phage’s lysing gene, leaving the phages in the lysogenic state, so that no progeny phages can be produced [143]. However, the antirepressor can inhibit cI activity, causing the phages to enter the lysis cycle and promote host cell lysis. The repressor was first identified in phage P22 and was named Ant [17]. A repressor named Qtip was found in phage VP882 [144]. Under the condition of high cell density, Qtip composed of 79 amino acids can recognize and bind the DNA-binding domain of the N-terminal of cI, inhibiting the activity of cI [145,146] (Figure 7b).

It is a common strategy for phages to resist the QS system to prevent AIs from binding to their receptors by producing receptors that can bind to AIs or by synthesizing proteins similar to AIs. For example, the gene *p37* that encodes the LuxR-type transcription factor was found in phage ΦARM81ld [18]. LuxR encoded by *p37* can bind to C4-HSL as AIs, which hinders the binding of C4-HSL and LuxR in the bacterial QS system. Phage VP882 encodes the VqmA QS receptor (VqmA_Phage_), a homology of Vibrio cholerae VqmA (VqmA_Vc_) [143]. VqmA_Phage_ can bind to 3,5-dimethylpyrazin-2-ol (DPO), which hinders the binding of VqmA_Vc_ and DPO (Figure 7c). Furthermore, the binding of VqmA_Phage_ to DPO can activate the expression of the anti-inhibitory factor Gp55. Gp55 can directly act on the cI repressor to inactivate cI, thus causing phages to enter the lysis cycle.

Phage DMS3 encodes Aqs1 protein that can bind to LasR, hindering the binding of *P. aeruginosa* AIs to its receptor LasR [147]. Aqs1, which consists of 69 residues, binds as a dimer to the N-terminal DNA-binding domain of LasR, hindering AHL binding to lasR in the lasR/lasI-type QS system. Aqs1 also inhibits the production of another class of AIs, the quinolone system (PQS), by down-regulating the pqsABCDE and phnAB operons [148] (Figure 7d). The quorum-sensing targeting protein encoded by phage LUZ19 can also inhibit PQS production by interacting with PQS biosynthetic pathway enzymes [149].

## 4. Conclusions

The mutual defense strategies between bacteria and phages are gradually being clarified. This review details bacterial anti-phage strategies and phages anti-defense measures to deeply understand the interaction mechanism between bacteria and phages, which is significance for the development and application of modern biotechnology.

At present, the defensive and anti-defensive measures between bacteria and phages have been able to solve many practical problems. For example, CRISPR-Cas technology can be used for pathogen nucleic acid detection [150], CRISPR-Cas gene editing technology for designing new strains with enhanced beneficial functions [151], the depolymerase produced by phages can be used as a new anti-biofilm agent [152], the problem of phage contamination during fermentation can be solved based on the interaction between phages and bacteria [153] and phage therapy can be used to treat bacteria, especially to treat infections with drug-resistant bacteria [154].

The emergence of superbugs poses a serious threat to human health, making the problem of drug resistance to bacteria the focus of global attention. Resistant genes can be transferred between bacteria to create new resistant combinations [155]. As a new type of therapy, phage therapy has been used in practical clinical treatment [156]. Phage therapy is expected to be used in combination with antibiotics to address the problem of bacterial resistance. One of the problems facing phage therapy is that bacteria evolve their resistance to phages [157]. The biofilm that forms when *P. aeruginosa* infects the lungs of people with cystic fibrosis can block antibiotics from entering bacterial cells [158]. At the same time, as a natural barrier, biofilm facilitates bacteria to stay on the surface of living and non-living organisms [159]. This can easily lead to nosocomial infections. Giant phages are often used to treat CF patients infected with *P. aeruginosa*. Phages can not only lysis bacteria but also reduce biofilms through polysaccharide depolymerase. However, the emergence of phage-resistant mutant strains prevents phage therapy from treating resistant bacteria. Understanding the evolution direction of phage-resistant mutant strains is important for the treatment of drug-resistant bacteria relying on cocktail therapy or the combination of phage therapy and antibiotics. Phage cocktail therapy, which consists of different phages, is more beneficial for the treatment of drug-resistant bacterial infections [160]. In conclusion, some of the defensive or anti-defensive systems found in the current research may be effective tools for solving practical problems in the future.

Among the defense and anti-defense measures between bacteria and phages, R-M, CRISPR-Cas, Abi and QS systems have been studied more deeply, while the anti-defense measures taken by phages against bacterial mutant receptors have been less studied. Besides, there are many new defensive and anti-defensive measures whose specific mechanisms of action have not been elucidated and need to be further studied.

The arms race between bacteria and phages is conducive to rapid coevolution between them. Meanwhile, temperate phages can also promote the adaptive evolution of the host [161]. Phages and their hosts can exchange genes through horizontal gene transfer, driving coevolution [162]. The coevolution of bacteria and phages can increase the mutation rate of bacteria. The mutant bacterial population can play a greater advantage in the arms race with phages [163]. At the same time, coevolution can also improve the rate of phage’s evolution [164]. However, the role of the community environment in the interaction between bacteria and phages is not fully understood, and many problems still need to be further explored [165].

## Figures and Tables

**Figure 1 ijms-24-04363-f001:**
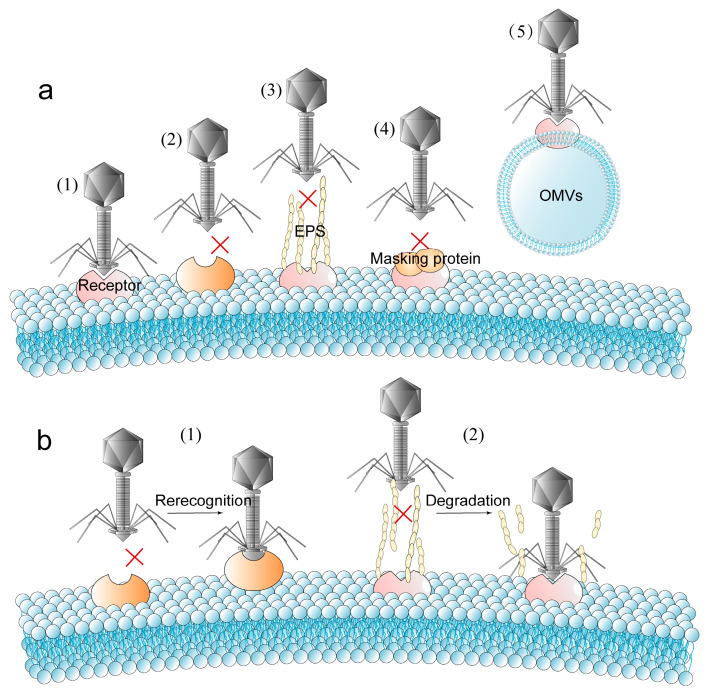
The arms race between bacteria and phages during the adsorption phase (the symbol red crosses in the figure indicate that the phage adsorption is blocked). (**a**) Under normal conditions, phages adsorb to host cells via bacterial surface receptors (1). Bacteria hinder the phage adsorption process by altering the receptor structure or by making the receptor unexpressed (2), by forming extracellular polymeric substances (EPS) (3), by masking proteins (4), or by masking the phage receptors by outer membrane vesicles (OMVs) with phage receptors (5). (**b**) The phages regain the ability to recognize and adsorb the host by recognizing new receptors through mutant receptor binding proteins (RBPs) (1) or by degrading EPS through polysaccharide depolymerases (2).

**Figure 2 ijms-24-04363-f002:**
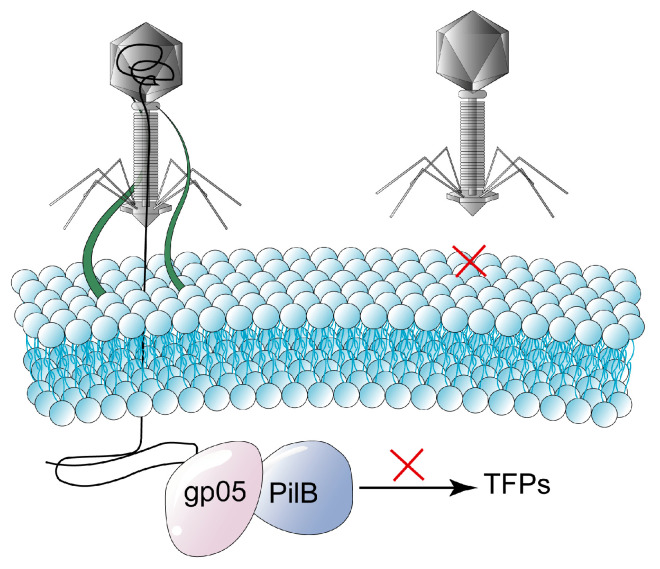
Bacteria hinder the phage’s DNA injection via superinfection exclusion (Sie) (the symbol red cross in the figure indicates that the adsorption of the phage is prevented). Phage D3112 encodes protein gp05, which inhibits the activity of PilB by interacting with PilB, affecting the assembly or function of TFPs and thereby preventing re-infection of host cells by phage MP22, which also uses *P. aeruginosa* TFPs as receptors.

**Figure 3 ijms-24-04363-f003:**
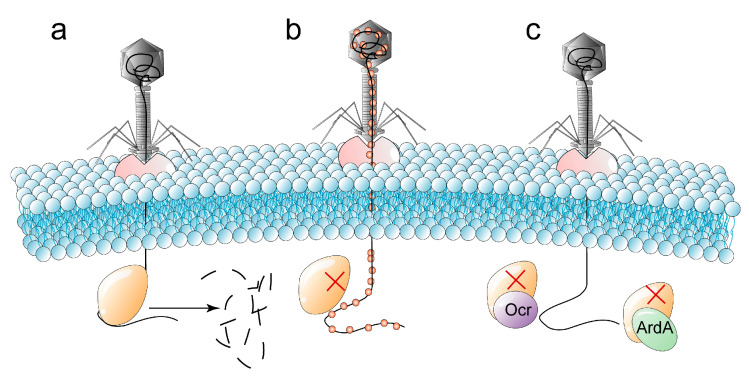
The arms race between bacteria and phages in terms of the restricting modification (R-M) systems. MTase methylates bacterial DNA recognition sites to distinguish unmodified exogenous DNA, and RNase cleaves the phosphodiester bond of unmethylated phage DNA, resulting in DNA breaks (the symbol red cross in the figure indicates that the cleavage of phage DNA by REase is blocked). (**a**) The unmethylated exogenous phage DNA is cleaved by REase, which hinders the replication process of the phages in the host cell. (**b**) The phages hinder the cleavage of the phage DNA by REase through modification of its genes. (**c**) Phages produce overcome the classical restriction (Ocr) and restriction of DNA A (ArdA), which can impede the cleavage by REase in the type I R-M system by imitating the shape and surface charge of phage DNA.

**Figure 4 ijms-24-04363-f004:**
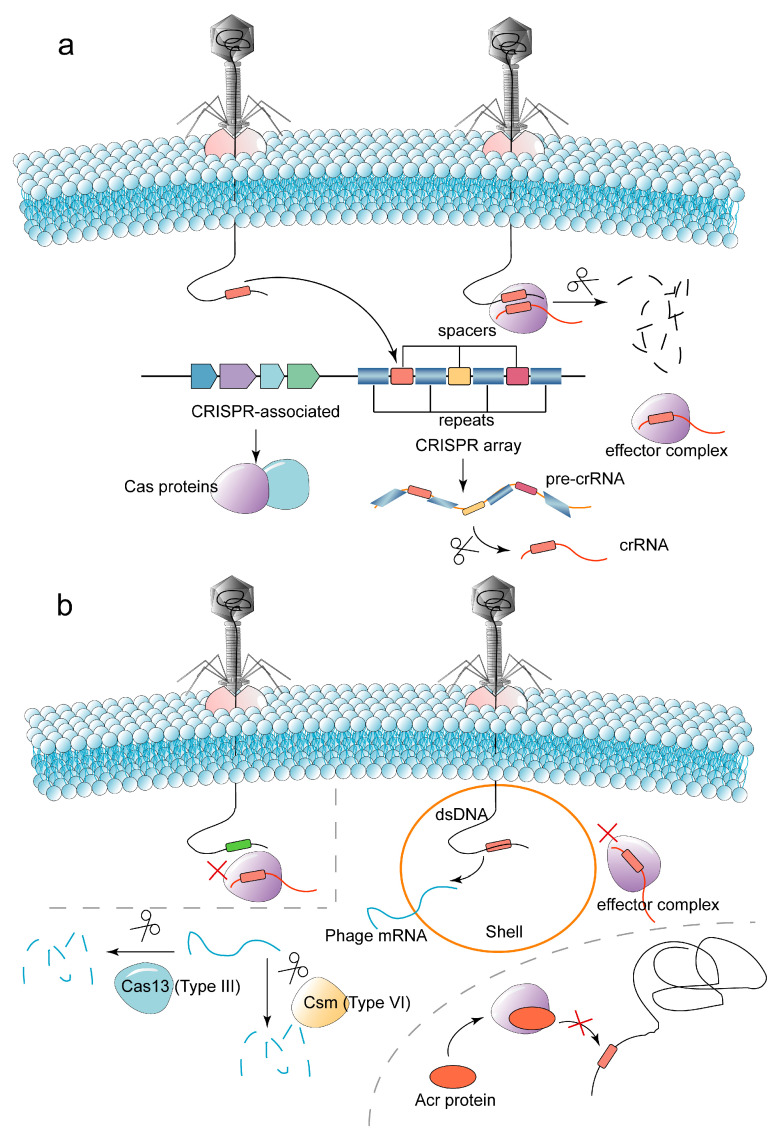
The arms race between bacteria and phages in terms of the CRISPR-Cas systems (the symbol red cross in the figure indicates that the recognition and cleavage of foreign genes by the effector complex is inhibited). (**a**) The CRISPR-Cas adaptive immune system consists of the CRISPR arrays and the *cas* genes. The effector complex of crRNA and Cas proteins targets and cleaves phage DNA/RNA. (**b**) Phages countermeasure against CRISPR-Cas, including mutating target sequences to block recognition of effector complexes; constructing nucleus-like compartments to shield nucleases; and expressing anti-CRISPR (Acr) to inhibit recognition of exogenous nucleic acids or impede cleavage of nucleic acids by Cas nucleases. Since phage mRNA expresses proteins in the bacterial matrix, phage mRNA can still be targeted and cleaved by type III and type VI CRISPR-Cas systems.

**Figure 5 ijms-24-04363-f005:**
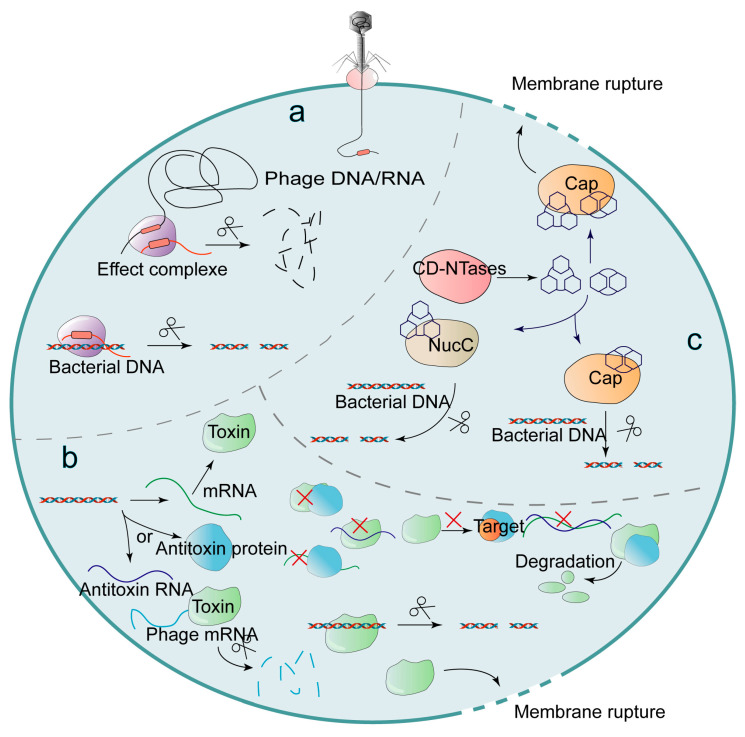
The abortive infection (Abi) systems are based on CRISPR-Cas, cyclic oligonucleotide-based anti-phage signaling system (CBASS) and toxin-antitoxin (TA). (**a**) The Cas proteins with endonuclease activity of CRISPR-Cas, which nonspecifically targets phage genes and bacterial DNA, is thought to be one of the reasons why CRISPR-Cas may mediate Abi in bacteria. However, the exact mechanism by which CRISPR-Cas causes Abi is not yet clear. (**b**) The TA systems are widely present in bacteria and consist of toxins and more unstable antitoxins (the symbol red cross in the figure indicates that the toxin is inhibited). Among them, toxins usually function as proteins to inhibit bacterial growth, and antitoxins function as proteins or RNAs to inhibit toxins. When phages infect bacteria, the TA transcriptional function of the bacteria is hindered and the unstable antitoxin is degraded before the toxin, leading to toxin accumulation. The toxin cleaves the mRNA of the phages. In addition, the toxin cleaves bacterial DNA and mediates the lysis of cell membranes, which leads to bacterial growth arrest or death. (**c**) The cyclic dinucleotide and cyclic trinucleotide synthesized by the cGAS/DncV-like nucleotide transferase (CD-NTase) in the CBASS immune system can promote bacterial membrane lysis or cleavage of bacterial DNA after binding to Cap proteins; the DNA endonuclease NucC in CBASS can cleave bacterial DNA after binding to cyclic AMP-AMP-AMP (cAAA), thus mediating Abi systems.

**Figure 6 ijms-24-04363-f006:**
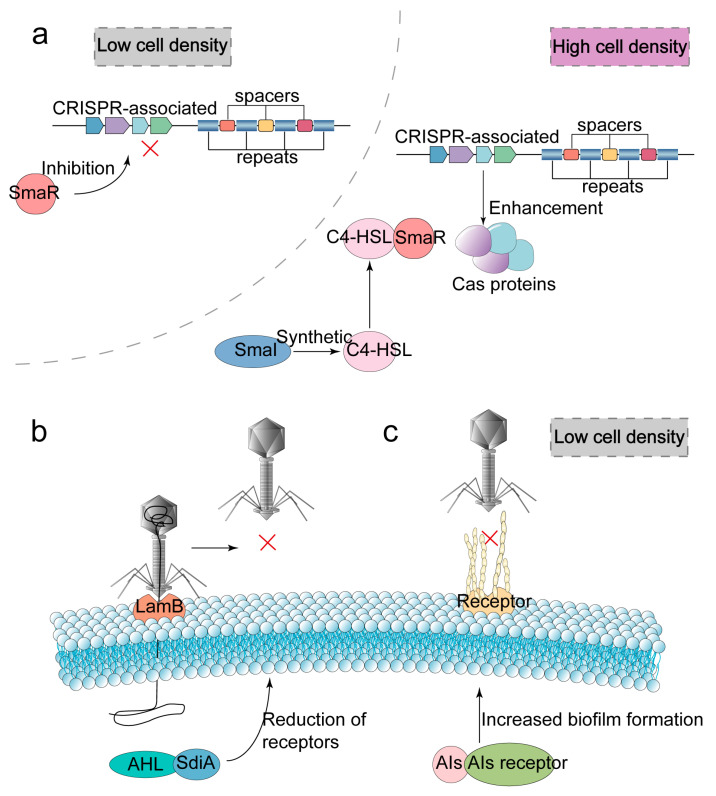
The QS system of bacteria resists phage infection by synergizing the CRISPR-Cas system or hindering phage adsorption. The QS system consists of QS signaling synthases, extracellular signaling molecules called autoinducers (AIs), and QS receptors. (**a**) Under conditions of high population density, the *N*-butanoyl-L-homoserine lactone (C4-HSL) produced by the smaI/smaR-type QS system binds to smaR, hinders the repressive effect of the DNA-binding repressor smaR on *cas* genes and promotes the expression of *cas* genes. (**b**) The binding of the QS signaling molecule acyl-homoserine lactones (AHL) to its receptor SdiA can reduce the number of λ phage receptors LamB (the symbol red cross in the figure indicates that the phage‘s adsorption is inhibited). (**c**) Under the condition of low cell density, the QS system can increase biofilm production to mask phage receptors and hinder phage adsorption.

**Figure 7 ijms-24-04363-f007:**
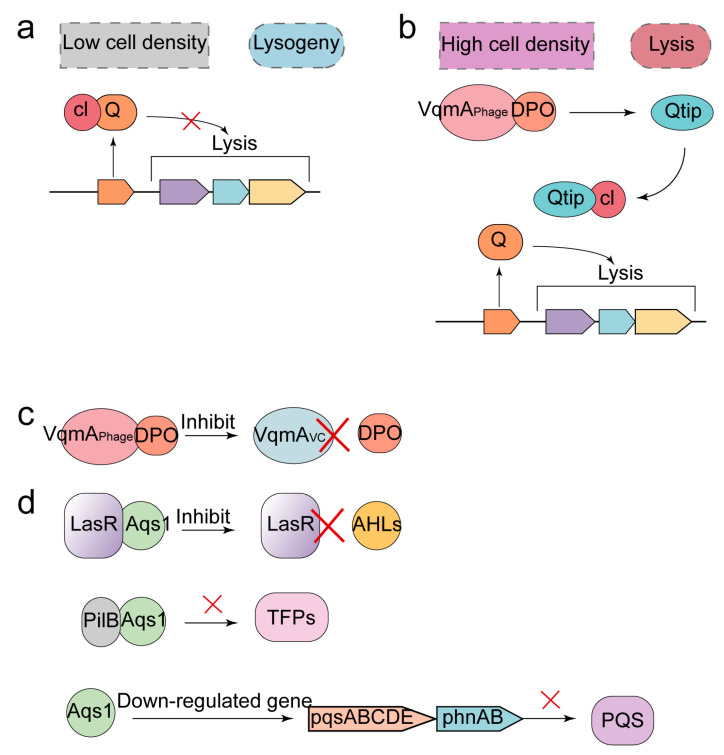
The hindering effect of phages on the QS system. (**a**) Under the low cell density conditions, the cI repressor binds to the Q promoter and inhibits the expression of phage lysis genes. (**b**) Under the high cell density conditions, Qtip recognizes and binds cI, thus inhibiting the binding of cI to the Q promoter and allowing the phages to enter the lysis cycle. The VqmA_Vc_ produced by Vibriophage VP882 is able to activate the expression of Qtip (Gp55) by binding to 3,5-dimethylpyrazin-2-ol (DPO). (**c**) Phages impede the binding of DPO to VqmA_Vc_ by encoding the receptor VqmA_Phage_, which can bind to DPO. (**d**) Phages synthesize the protein Aqs1, which resembles AHLs, thereby impeding the binding of AHLs to the QS receptor LasR. In addition, by interacting with PilB, AHLs impede the adsorption of phages to *P. aeruginosa* Aqs1 can downregulate the pqsABCDE and phnAB operons and inhibits the production of the quinolone system (PQS).

**Table 1 ijms-24-04363-t001:** Defense and anti-defense strategies between bacteria and phages.

**Bacterial Anti-Phage Strategies**	**Description of the Strategies**
Blocking phage adsorption	Loss or structural change of receptors.
	Mask phage receptors by physical barriers such as outer membrane vesicles (OMVs), masking proteins or extracellular polymeric substances (EPS).
Blocking phage injection	Avoid bacterial re-infestation by another identical or highly similar phage after infection by a lysogenic phage through superinfection exclusion (Sie).
Interfering with phage replication	Cleavage of phosphodiester bonds of unmethylated phage DNA by the restricting modification (R-M) systems.
	Recognition and cleavage of phage DNA/RNA by the effector complex consisting of Cas protein and guide RNA (gRNA) of CRISPR-Cas system.
	Bacteria inhibit their metabolism through abortive infection (Abi) leading to their own growth arrest or death, thus avoiding the maturation and release of phages.
Quorum sensing (QS)	Synergistic CRISPR-Cas systems stimulate the expression of *cas* genes and promote recognition and cleavage of target genes under conditions of high population density.
	Reduce the expression of phage receptors or increase biofilm production to mask phage receptors under conditions of low cell density.
**Anti-defense strategies of phages**	**Description of the strategies**
Regaining the ability to identify and adsorb host	Mutate the receptor binding proteins (RBPs).
	Degrade EPS by depolymerase.
Anti-defense strategies for R-M systems	Modify the genes of phages. Evolve to overcome classical restriction (Ocr) protein and the restriction of DNA A (ArdA) protein that can inhibit the R-M complex.
Anti-defense strategies for CRISPR-Cas systems	Construct nucleus-like compartments to shield nuclease.
	Mutate the target sequence to block the recognition of effector complexes.
	Through anti-CRISPR (Acr) protein to inhibit recognition or cleaving of foreign nucleic acids by effector complexes.
Anti-defense strategies for QS system	Expression of anti-repressor that can bind cI repressor, thereby allowing the phage to enter the lysis cycle.

## Data Availability

Not applicable.

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
