# Peer review of "Unveil the Secret of the Bacteria and Phage Arms Race"

_ijms, 2023, doi:10.3390/ijms24054363_

Round 1

Reviewer 1 Report

Review of ‘unveil the secret of bacteria and phages arm race’

In this review article the authors discuss phage and bacterial defence mechanism. The authors have written the article thoughtfully, presented their points clearly and all figures are excellent. Overall the article is excellent and will be a valuable article for the phage community.

My general comments are the article needs to be proof-read as there a few errors and the authors keep jumping from ‘bacteriophages’ to ‘phages’ in the manuscript.

Minor comments are:

Abstract

·       Line 9 – there is a missing word in ‘phages from adsorbed’. Change this to ‘phages from being adsorbed’

1.     Introduction

·       Line 27 – in brackets after ‘bacteriophages’ add ‘phages for short’ as in the next line you refer to them as phages

·       Lines 33 to 36 – include references

·       Line 49 – delete full stop

·       Line 51 – here you refer to them as bacteriophages but in previous paragraph you referred to them as phages. Pick one and stick with it throughout the manuscript

·       In Table 1 what do you mean by ‘specific ways’? It might be better rephrased as description on strategy. Overall the whole table is unclear and difficult to follow, I suggest the authors re-structure the table to make it easier to follow – maybe add bullet points to the ‘specific ways’ column, make the strategies sub-headings bold   

2.     Bacterial anti-phage strategies

·       Lines 58 to 60 – remove these sentences as I think you have accidently included the text in the manuscript template  

·       Lines 60 to 64 are repeat sentences from the introduction, please remove

·       The section introduction (lines 57 to 70) needs to be re-written as the current text is not linked to this section

·       Line 74 – delete word ‘ roughly’

·       Within figure 1 you mention left 1, right 2…. It would make is easier to follow if you include these numbers within the figure. Aside from this the figure is excellent  

·       Line 117 – italicise ‘A. baumannii’

·       Line 457 – again remove the word ‘roughly’ and the word ‘kinds’

3.     Anti-defence strategies of phages

·      Line 597 – delete word ‘roughly’

·      Line 633 - italicise ‘P. aeruginosa’

4.     Conclusions

·      Line 686 – what do you mean by ‘deepening’

Author Response

Reviewer #1

In this review article the authors discuss phage and bacterial defense mechanism. The authors have written the article thoughtfully, presented their points clearly and all figures are excellent. Overall the article is excellent and will be a valuable article for the phage community.

Response: Thank you very much for taking the time to review this review, and thank you for your recognition of this review.

My general comments are the article needs to be proof-read as there a few errors and the authors keep jumping from ‘bacteriophages’ to ‘phages’ in the manuscript.

Response: Thank you for your valuable suggestions. As you suggested, we have added ‘phages for short’ in brackets after ‘bacteriophages’, and use ‘phages’ consistently in the later text.

Minor comments are:

Abstract

  • Line 9 – there is a missing word in ‘phages from adsorbed’. Change this to ‘phages from being adsorbed’

Response: Thank you for your comments, we have modified this. (Line 9)

  1. Introduction
  • Line 27 – in brackets after ‘bacteriophages’ add ‘phages for short’ as in the next line you refer to them as phages

Response: Thank you for your valuable advice, we have modified this point according to your suggestion. (Line 27)

  • Lines 33 to 36 – include references

Response: Thank you for your advice, we have added new references to this section, “After infecting the host, temperate phages do not lysis the host cell but remain in the lysogenic state, integrating the phages genome into the host genome, becoming prophage and replicating with the host replication. Lytic phages can cause lysis and death of the host [2]. The process of lytic phages infecting the host includes five parts: adsorption, injection, biosynthesis, assembly and release. Under certain conditions, temperate phages can enter the lysis cycle from the lysogenic state, causing the host cell to the lysis and die [3].” (Line 31 to 37)

  • Line 49 – delete full stop

Response: Thank you for pointing out this error, we have removed the full stop. (Line 49)

  • Line 51 – here you refer to them as bacteriophages but in previous paragraph you referred to them as phages. Pick one and stick with it throughout the manuscript

Response: Thank you for your suggestion. We have revised the manuscript as you suggested and changed all parts to ‘phages’ except for the first time to ‘bacteriophages’.

  • In Table 1 what do you mean by ‘specific ways’? It might be better rephrased as description on strategy. Overall the whole table is unclear and difficult to follow, I suggest the authors re-structure the table to make it easier to follow – maybe add bullet points to the ‘specific ways’ column, make the strategies sub-headings bold   

Response: Thank you for your sincere advice. We have replaced the original ‘specific ways’ with ‘Description of the strategies’. To make the table easier to understand, we have changed the table from four columns to two columns and bolded the sub-headings.

The table has been changed as shown below.

Table 1. Defense and anti-defense strategies between bacteria and phages.

Bacterial anti-phage strategies

Description of the strategies

Blocking phage adsorption

Loss or structural change of receptors.

Mask phage receptors by physical barriers such as outer membrane vesicles (OMVs), masking proteins or extracellular polymeric substances (EPS).

Blocking phage injection

Avoid bacterial re-infestation by another identical or highly similar phage after infection by a lysogenic phage with superinfection exclusion (Sie).

Interfering with phage replication

Cleavage of phosphodiester bonds of unmethylated phage DNA by the restricting modification (R-M) systems.

Recognition and cleavage of phage DNA/RNA by the effector complex consisting of Cas protein and gRNA of CRISPR-Cas system.

Bacteria inhibit their metabolism through abortive infection (Abi) leading to their own growth arrest or death, thus avoiding the maturation and release of phages.

Quorum sensing (QS)

Synergistic CRISPR-Cas systems stimulate the expression of cas genes and promote recognition and cleavage of target genes under conditions of high population density.

Reducing the expression of phage receptors; or increasing biofilm production to mask phage receptors under conditions of low cell density.

Anti-defense strategies of phages

Description of the strategies

Regaining the ability to identify and adsorb host

Mutate the receptor binding proteins (RBPs).

Degrade EPS by depolymerase.

Anti-defense strategies for R-M systems

Modify the genes of phages.

Evolve to overcome classical restriction (Ocr) protein and the restriction of DNA A (ArdA) protein that can inhibit the R-M complex.

Anti-defense strategies for CRISPR-Cas systems

Construct nucleus-like compartments to shield nuclease.

Mutate the target sequence to block the recognition of effector complexes.

Through anti-CRISPR (Acr) protein to inhibit recognition or cleaving of foreign nucleic acids by effector complexes.

Anti-defense strategies for QS system

Expression of anti-repressor that can bind cI repressor, thereby allowing the phage to enter the lysis cycle.

  1. Bacterial anti-phage strategies
  • Lines 58 to 60 – remove these sentences as I think you have accidently included the text in the manuscript template  

Response: We are very sorry that we accidentally added these sentences from the manuscript template to this manuscript, and we have removed them in the revised manuscript.

  • Lines 60 to 64 are repeat sentences from the introduction, please remove

Response: Thank you very much for your suggestion, we have removed these duplicate sentences.

  • The section introduction (lines 57 to 70) needs to be re-written as the current text is not linked to this section

Response: Thank you very much for your suggestion. In order to harmonize with the format of Part ‘3. Anti-defense strategies of phages’ and to avoid repetitive ideas in the manuscript, we have deleted these words in lines 57-70. Thanks again for your advice.

  • Line 74 – delete word ‘roughly’

Response: Thank you for your suggestion, we have removed ‘roughly’. (Line 72)

  • Within figure 1 you mention left 1, right 2…. It would make is easier to follow if you include these numbers within the figure. Aside from this the figure is excellent  

Response: Thank you for your suggestion on figure 1. We have modified the figure 1 as shown below and changed the figure legend accordingly.

And thank you for your compliments of the figure 1.

Figure 1. The arms race between bacteria and phages during the adsorption phase. (a) Under normal conditions, phages adsorb to host cells via bacterial surface receptors (1). Bacteria hinder the phages adsorption process by altering the receptor structure or by making the receptor unexpressed (2), by forming extracellular polymeric substances (EPS) (3), by masking proteins (4), or by masking the phage receptors by outer membrane vesicles (OMVs) with phage receptors (5). (b) The phages regain the ability to recognize and adsorb the host by recognizing new receptors through mutant receptor binding proteins (RBPs) (1) or by degrading extracellular polymers through polysaccharide depolymerases (2).

  • Line 117 – italicise ‘A. baumannii’

Response: Thanks for the suggestion, the ‘A. baumannii’ has been changed to italic. (Line 116)

  • Line 457 – again remove the word ‘roughly’ and the word ‘kinds’

 Response: Thank you for your suggestion, we have removed ‘roughly’ and ‘kinds’. (Line 457)

  1. Anti-defence strategies of phages
  • Line 597 – delete word ‘roughly’

Response: Thank you for your suggestion, we have removed ‘roughly’. (Line 597)

  • Line 633 - italicise ‘P. aeruginosa’

Response: Thanks for the suggestion, the ‘P. aeruginosa’ has been changed to italic. (Line 633)

  1. Conclusions
  • Line 686 – what do you mean by ‘deepening’

Response: Thank you for your suggestion. The sentence ‘with the deepening of the research on bacteria and phages, the interaction between them has been gradually clarified’ has been rewritten as ‘the mutual defense strategies between bacteria and phages are gradually being clarified’. (Line 686 to 687)

Reviewer 2 Report

This is a useful and timely review that gives a good overview of the evolving strategies of phage and bacteria. While there are no real insights here the article brings the literature together. It does lack detail and can be superficial in parts. The authors should add some commentary on the discovery of CAS-CRIPSR in Str thermophilus by Barrangou et al. 2007 and the significance of this system in lactic starter cultures and the evolution of new phages. The recent review by Avery Roberts and Rodolphe Barrangou (2020) is missing. I recommend that the authors read this paper and make appropriate adjustments to the text.

The authors should remove superlatives and try and any non-scientific language.

I enjoyed reading the review.

The English use is sufficient to understand the points made by the authors but it could be further improved.

Author Response

Reviewer #2

This is a useful and timely review that gives a good overview of the evolving strategies of phage and bacteria. While there are no real insights here the article brings the literature together. It does lack detail and can be superficial in parts. The authors should add some commentary on the discovery of CAS-CRIPSR in Str thermophilus by Barrangou et al. 2007 and the significance of this system in lactic starter cultures and the evolution of new phages. The recent review by Avery Roberts and Rodolphe Barrangou (2020) is missing. I recommend that the authors read this paper and make appropriate adjustments to the text.

The authors should remove superlatives and try and any non-scientific language.

I enjoyed reading the review.

The English use is sufficient to understand the points made by the authors but it could be further improved.

Response: Thank you so much for your insightful advice. The bacterial and phages arms race involves many strategies, and we have discarded the “details” in order to try to cover all defense and anti-defense strategies.

We have carefully read the article and review you recommended and added them to the manuscript, “And the higher the content of CRISPR spacers, the more sensitive the bacteria to phages [64].” (Line 305 to 306), “CRISPR-Cas gene editing technology for designing new strains with enhanced beneficial functions [152]” (Line 694 to 695), both of which are very informative. In particular, the part of your recommended review on designing new strains based on CRISPR-Cas gene editing technology is very inspiring to us. This involves the application of CRISPR-Cas technology, and the purpose of our manuscript is precisely to allow researchers to understand the strategies of the arms race between bacteria and phages and to use these strategies to solve practical problems. In addition, this review is more detailed than our manuscript regarding the classification of CRISPR-Cas. Thank you for your recommendation.

We have removed all non-scientific language such as “roughly”.

Thank you for your advice and your positive comments on this manuscript.

 As non-native speakers, we tried to polish the English of this manuscript using a website or software.
